# The Disruptive Adaptations of Construction 4.0 and Industry 4.0 as a Pathway to a Sustainable Innovation and Inclusive Industrial Technological Development

**Amusan Lekan [1],\* , Aigbavboa Clinton [2],\* and James Owolabi [1],\***

[1] Building Technology Department, Covenant University, Canaan Land, PMB1023 Ota, Ogun State, Nigeria
[2] Cidb Centre of Excellence & Sustainable Human Settlement and Construction Research Centre, Department of Construction Management & Quantity Surveying, Faculty of Engineering and the Built Environment, Doorfoiten Campus, University of Johannesburg, Maropeng 198, South Africa
\* Correspondence: lekan.amusan@covenantuniversity.edu.ng (A.L.); caigbavboa@uj.ac.za (A.C.); james.owolabi@covenantuniversity.edu.ng (J.O.); Tel.: +234-803-074-3025 (A.L.)

**Abstract:** Construction 4.0 (C4.0) has tremendously impacted construction activities worldwide in recent times. This effect was made possible on account of innovations brought about by Industry 4.0 (I4.0). Industry 4.0 has the potential to create Construction 4.0 through the integration of the design, construction and maintenance of infrastructure through useful component integration for industrial and technological development. Therefore, this study aimed to present a pathway for achieving sustainable innovations and inclusive technological and infrastructural developments. The following parameters were reviewed in this study as part of the goals and objectives set in the survey: identifying the adaptable areas of Construction 4.0 in design, planning, construction and maintenance as part of infrastructural innovation in order to study the industrial application drivers of I4.0 and C4.0 hindrances in achieving C4.0; achieving the automation dream through C4.0, benchmarking the social and economic implications of C4.0 and identifying the issues and challenges in achieving sustainable innovation through infrastructural development and documenting the disruptive tools of C4.0 in achieving a sustainable design through technological development and examining the critical factors influencing the effective adaptation of C4.0 in achieving growth. The authors utilised 200 construction firms for this study using the Cochran and Slovin's formulas. In addition, the sample size of 150 respondents that constituted the study were construction professionals. The respondents used the simple percentage, relative index, Spearman's rank, Mann–Whitney U test, Kendall's Tau test, Student's *t*-test, ANOVA and chi-square tools in the data processing. The study found out, among other things, the following as part of the parameters earlier proposed: the introduction of a circular economy by adopting intelligent innovation, engaging new tools, technological innovation diffusion and the vertical and horizontal integration of versatile tools like I4.0 and C4.0 for inclusive technological development. This study recommended the objective and effective adaptation of I4.0 tools to enhance C4.0 for technical development, circular economic integration and a framework for sustainable innovation and a system for the inclusive monitoring of innovations in the design and planning of construction maintenance.

**Keywords:** construction; innovation; adaptation; technology; sustainability; inclusion

## 1. Introduction

The construction industry has experienced tremendous changes in recent times.
This sector has received intense attention in the wake of the fulfilment of various construction-related components of the United Nations' Development Goals [1–3].
These goals were accorded the utmost priority, since they touch on human welfare, health, sustenance, shelter, wellbeing and development. Additionally, the components

of the plans that include provisions for infrastructure, industrial development and technological advancement have received a significant boost; more than ever before, there is a call to provide affordable facilities. A facility that is resistant to the wear and tear often associated with prolonged usage would be a self-sustaining and maintenance-free facility, one with renewable components. Based on users' experiences, facility needs are some of the reasons that warrant a paradigm shift in the direction of building and facility automation for facilities that would be self-sufficient, sustainable and cost-efficient. Cost efficiency in building and construction works begins from conceptualising at the idea stage and includes the post-occupation stage. This is referred to as the building's life cycle, a cycle of events that originates at the conception of an idea stage until the post-occupancy location. Cost-effectiveness on a project starts from forming an idea exclusively through brainstorming or the Delphi approach. However, the actual construction costs are unknown at this stage. The cost expert often leverages their intuition to conceptualise the project component costs.

Similarly, the design and cost stages have their limitations, which encompass inefficient designs and cost systems. The above fact necessitates the evolution of a smart system that could combine efficiency with cost-effectiveness in design and construction. Construction work is multicomponent, with the individual components requiring effective synchronisation for corporate success. The problem often encountered stems from the non-effective synchronisation of the parts to non-judicious resource allocation, which has created a call for concern among construction stakeholders, who always want their value for the money they invest in a business. The authors of [1] affirmed that, for attaining an effective deployment of innovation in the construction sector, there is a need for an inclusive process that considers the end-users' perspectives and post-occupation needs. The future of a project lies in what occurs at the post-occupation stage. Therefore, an inclusive approach, innovation and the technological components that take care of the facility users' immediate and future needs are essential [2,3]. Technological developments have effectively integrated project components that help predict a project's future from present to end. Therefore, technological development needs a sustainable system to consider the current system's provisions without compromising the future. According to the authors of [3], a sustainable system is possible by continuously innovating the best strategy and approach to an existing design. Many practical innovations have been introduced for problem-solving in the construction industry. Therefore, the critical aspects of construction and post-occupation management need thoroughly integrated, which requires the holistic inclusion of the relevant industrial application components necessary for this study [4–8]. The characteristics of Industry 4.0 were blended with Construction 4.0 to achieve inclusive technological development [9–12].

Therefore, this study aims to achieve an inclusive pathway towards the sustainable innovation and inclusive technological developments that are necessary to support sustainable design and inclusive industrial development goals 9 and 11 [13–15]. The objectives of goals 9 and 11 include achieving sustainable infrastructure, resilient structures and inclusive technological developments [16–19]. The goals and the mechanisms of their expression form the seeds of the objectives and will help to identify gaps in them after reviewing the relevant literature; the purpose of this is presented in Section 1.1.2.

There is a need to delineate the goals 9 and 11 of the United Nations Development Programme (UNDP) Goal 9 of the UNDP is about industry innovations and infrastructural provisions. According to [19,20], any nation's economy needs drivers to facilitate the rapid economic growth and development embedded in UNDP's goal, which includes shelter and infrastructure provisions. Therefore, the UNDP has taken a proactive approach toward preempting the imminent shelter shortage being experienced worldwide. Investing in scientific research would make a difference in the ever-growing need for energy transportation and information communication technology. Therefore, the authors of [19,21] submitted that technological advancements and sophistication are required for innovative solutions and industrial development in order to provide infrastructures.

Similarly, goal 11 addresses sustainable cities and communities. The way cities are built and designed is a significant component. The authors of [12] submitted that the ever-growing population has led to the emergence of megacities. Innovations and technological developments need to be preserved for sustainability. The inventions and industrial developments encapsulated in goal 9 provide the catalyst for the infrastructural developments required to provide affordable and sustainable cities, as prescribed in goals 9 and 11. This tends to bridge the technological divide affecting the provisions of sustainable cities and infrastructures.

The remaining part of this study is structured as follows: A literature search was carried out and is presented in Section 2.1, containing the parameters that include Construction 4.0 (C4.0), sustainable development goals 9 and 11, the concept of technological inclusion, the factors influencing the achievement of Industry 4.0 (I4.0) and C4.0 and innovations for technological developments. As presented in Section 3, a survey research approach was adopted for this study. The authors utilised a population of 200 construction firms. Eventually, a sample size of 150 respondents was used by the authors for the study. The respondents comprised selected professionals from construction firms. The authors calibrated the implemented questionnaire as a Likert scale of 1–5 for the numerical and ordinal data. The authors used statistical tools to process the collated data, such as simple percentage, the relative index, ANOVA, chi-square, a homogeneity test, Spearman's rank, the Mann–Whitney *U* test and others [21–23]. The results are presented in the tables and charts. The other sections of the article include the section presenting the results, the discussion section, the conclusion, the recommendations and the reference section.

### 1.1. Aim and Objectives of the Study

#### 1.1.1. Aim of the Study

This study aims to present a pathway for achieving sustainable innovations and inclusive technological and infrastructural development using Construction 4.0 and Industry 4.0 attributes.

#### 1.1.2. Objectives of the Study

The following items were reviewed in this study as part of the set objectives. They were to:

i.      identify the areas of Construction 4.0 (C4.0) that can be adopted for improvement in infrastructural innovation in design, planning, construction and maintenance [16,17].

Analytical tools: Relative Index, Mean Index and Pearson's Spearman's Rank.

ii.     study the industrial application drivers of I4.0 and C4.0 and the hindrances in achieving C4.0 from the professional's perspective [17,20].

Analytical tools: Relative Index, Mean Index, Pearson's Spearman's Rank, ANOVA and Mann–Whitney *U* Test.

iii.    identify strategies that could be used to achieve an inclusive industrial automation development dream through C4.0 [9,15,21].

Analytical tools: Relative Index, Mean Index, Pearson's Spearman's Rank, Chi-square and Friedman's ANOVA.

iv.     study the identified social and economic implications of the C4.0 innovations in industrial development [21,23].

Analytical tools: Relative Index, Mean Index and Pearson's Spearman's Rank.

v.      identify issues and challenges involved in achieving a sustainable design in infrastructural development using C4.0 [22,24].

Analytical tools: Relative Index, Mean Index and Pearson's Spearman's Rank.

vi.     study the disruptive tools of C4.0 that are suitable for achieving inclusive and sustainable innovation for technological development [10,16,24].

Analytical tools: Relative Index, Mean Index and Pearson's Spearman's Rank.

vii.  profile critical factors that influence the practical adaptation of C4.0 in achieving industrial development [10,16,17,25].

Analytical tools: Relative Index, Mean Index and Pearson's Spearman's Rank.

viii.  present a pathway for Construction 4.0 and Industry 4.0 for sustainable innovation and inclusive technological development [10,16,23,26,27].

Analytical tools: Relative Index, Mean Index, Pearson's Spearman's Rank and Mann–Whitney U Test.

Hypotheses for the Study

Some hypotheses are articulated for the operationalization of the objectives. The views are analysed for validity and relevance and to further explore the contents of the objectives.

**Hypothesis 1. (Objective 2).** *There is no significant difference in the opinions of professionals on the rating of the drivers of I4.0 and C4.0.*

*There is a significant difference in professionals' opinion on the rating of the drivers of I4.0 and C4.0.*

**Hypothesis 2. (Objective 2).** *There is a considerable difference in the rating of the drivers and hindrances in achieving C4.0 from the professional's perspective.*

*There is no considerable difference in the drivers and hindrances ratings in achieving C4.0 from the professional's perspective.*

**Hypothesis 3. (Objective 7).** *There is no uniformity of opinion on the disruptive tools of C4.0 adaptation in achieving sustainable technological development.*

*There is uniformity of opinion as regards to disruptive tools of C4.0 adaptation in achieving sustainable technological development.*

**Hypothesis 4. (Objective 6).** *There is no agreement on the ratings of effective practical adaptations of C4.0 in technological development.*

*There is agreement on the ratings of effective practical adaptations of C4.0 in technological development.*

## 2. Review of Related Literature

This section presents some concepts that help the authors understand the research questions' parts and objectives set in Section 1 of this study. The unit covers the study areas such as understanding the I4.0 and C4.0 concepts, the concept of C4.0, the sustainable development (SDG) goals 9 and 11 in perspective, Construction 4.0 adaptation for technological development, the concept of technological inclusions and the factors influencing the achievement of Industry 4.0, among others.

### 2.1. Understanding the I4.0 and C4.0 Concepts

There is a robust connection between C4.0 and I4.0. C4.0 has been observed over time as significant drivers of economic development. It embodies industrial development through the technology of Industry 4.0 [I4.0]. There is a need for a complete system that is sustainable to consider providing the current system without comprising the future. According to [7], a sustainable system is possible by continuously innovating the best strategy and approach to the existing system. In [8], it was stated that many innovations have come onboard in the form of applications targeted towards problem-solving in the construction industry. Innovation applications help carry out feasibility studies, design calibration, human resources profiling, cost scheduling and mapping, construction devices, maintenance and post-occupancy resources monitoring. However, the question is, how sustainable are the innovations? Is there a continuity of design and concept? Is the application formidable enough to ensure consistency? How effective is the cost of creation, procurement deployment and monitoring? The questions raised are some of the lines of thoughts that inspired this research work. Achieving an inclusive and sustainable industrial development that would help create a significant technological development

requires cutting-edge innovation and techniques, which is obtainable through I4.0 and C4.0 [9–11].

Construction 4.0 (C4.0) and Industry 4.0 (I4.0) were described by [12,13] as the proliferation of state-of-the-art equipment and tools in carrying out operations predominantly being carried out in the past engaging old methods. It involves the deployment and application of digital technology to integrate the functional components of construction operation and processes. The authors of [14] posited that the venture that introduces automation into the construction process functionality is good. Since the introduction to the construction arena, it has led to an enhanced construction output of machine, workers and professionals. Construction 4.0, as submitted by [15], involves the unilateral integration of processes involved in construction activities using digital automation.

### 2.2. Sustainable Development (SDG) Goal 9 and 11 in Perspective

The concept of sustainability is an interesting one that addresses the future from the present. Sustainability is an issue for all aspects of human endeavours, and therefore, multidisciplinary. The authors of [16] submitted in a work that sustainable goals 9 and 11 are about discovering a dynamic way of creating a formidable and renewable future. Sustainable development goal nine would ensure the fulfilment of industrial development. The authors of [17–19] described the components of sustainable goals 9 and 11 as an axiom to fulfil infrastructural development, innovation articulation and industry development. Infrastructure is regarded as capable of providing a slum- and shambles-free society. Everyone has the opportunity to access resilient infrastructure, resilient materials, road infrastructure and building facilities. Inclusive and sustainable industrialisation is another cardinal point of the goal 9 and 11 specifications. There is a need for an inclusive component for rationalising the structures and frameworks for technological development to create an ambient environment for industrial development. A submission in [20] stated that sustainable development goals 9 and 11 were stated to address three (3) important aspects of development that touch on sustainable development, including industrialisation, infrastructural development and innovation. Industrialisation enables the complete development of innovation and the introduction of innovative skill and technological transfer. Industrialisation makes possible technological growth and development and inclusive societal growth. According to [20], it enables the fusion of urban and rural communities towards inclusive growth. It bridges the gap between giant and small-scale enterprises, thereby bringing more income to local enterprises. The view expressed above by [20] relates to the rationalisation of an economic issue surrounding society. The authors of [21] described the scenario as an economic apparatus to achieve sustainability. The study expressed sustainability as encapsulated in the Millennium Development Goal (MDG) goals as resting on three pillars, including an ecological pillar, the social pillar and economic pillar [22,23]. The ecological perspective enables the societal component to grow independently and, for the ecological component, to maintaining their functions [13,15]. Economic sustainability is when a system is allowed to settle the wellbeing of society over time. It works with framework development. However, in this study, sustainability is viewed in line with the requirements of goals nine and eleven to meet the technological need. The infrastructural composition provides inclusive society and technology for a robust industrial development, as supported in [10,11,13].

### 2.3. Construction 4.0 Adaptation for Technological Development

The authors viewed the adaptation of C4.0 from the perspective of functionality. The functionality was based on the components that C4.0 impacts. The essential construction elements that C4.0 impacts include design, planning, procurement, construction, maintenance and post-occupation management. In design and planning, there is automation that assists in getting work done. For instance, in a comfortable and orderly manner, the old ways of carrying out planning and the brainstorming Delphi approach are predominant, especially since the Delphi approach is subjective. Results in those days depend upon the

opinion of the brainstorming and planning panel. The authors of [23] submitted that the advent of operations research has brought up inventions to manage different construction works. Decision alternatives in planning materials and human resources onsite have helped professionals in carrying out their tasks. The other options come in the form of software that provides some of the automation sources adaptable in construction attributed to C4.0 inventions. In Constructor 2019, [24] digital tools that enabled functional adaptation in the construction process were articulated. It spanned from design digital tools, telematics, artificial intelligence, augmented knowledge, virtual reality and parametric modelling.

In design, the 3D design and printing method has replaced the old 2D system and has changed the design and architectural masterpiece landscape.

Telematics also presents a vital adaptation area of C4.0. Telematics combines the instrumentality of construction informatics with telecommunications. Telecommunication systems and information systems are fused into a typical telematics software loaded onto machines and a connected plant and the equipment on site. This view is supported in [8–10]. In the mentioned authors' submission, i.e., [8,13,16], laser equipment and the Global Positioning System (GPS) position system have been incorporated into construction equipment in recent times. The GPS record movements and positions and presents computerised maps well-situated on grid lines for easy decoding by supervisors and decoding systems.

Similarly, [17] documented the application of artificial intelligence that mimics human [19,20] intelligence, which works on computer programmes and software; this fact, according to [12,13], has led to the new field of study referred to as Biomimetic. Augmented and virtual reality as one of the adaptable areas in C4.0 has led to the emergence of system performance simulators using a virtual existence method.

Parametric modelling and point cloud technology have penetrated the construction arena in developed industrialised countries as one of the adaptations of C4.0. For instance, point cloud technology and parametric modelling technology were integrated in the Korea, Japan and Singapore construction sectors. These venture have led to productivity enhancement. According to [6,7], the system works based on a collection of a data points in space as generated through 3D technology parametric models and then blending the data into an appropriate model that fits the collated data into suitable logic. The laser scanner equipped and mounted on the application helps capture the as-built shape of an object for precision measurement and calibration.

*2.4. Concept of Technological Inclusions*

Technological inclusion is necessary for the construction industry and technological development. Inclusion involves technical integration [18], alluding to the fact that technology integration in a learning system involves incorporating different components that assist learning. The concept of technological inclusion teaches how to effectively leverage and integrate technology and innovation tools into construction works. The inclusion area includes a mobile learning system, electric learning system and digital learning system [21]. Technological inclusion is achievable through two approaches dealing with a technical-focused system and the means of technology delivery. An integrated system is necessary to design, construct, maintain and monitor to achieve an inclusive technological society. It is needed to galvanise the component of technology being brought onboard. Different methods are often applicable to creating specialised development plans. The plans are referred to as technical learning pedagogy, according to [13]; the methods include electronic transfer mode, mobile transfer mode, digital transfer mode, collaborative transfer mode, active transfer mode and blended transfer mode.

In the active transfer mode, the experience garnered according to [16,26,27], the professionals and workers learn from one another and share their experiences. Similarly, in blended technology, transfer modes and different applications are used to integrate functions that enable all aspects of the construction process to be adequately integrated. Such applications that assist in this regard include tablets, I-pods, podcasts, Android phones, mobiles and other digital devices. The authors of [16] submitted that the applications are

some of the interfaces through which inclusive technologies are deployed in the construction industry. In recent times, the paradigm of technological inclusion has shifted in the direction of flipped technology, as presented in this study. Some plants and machines are highly fortified with state-of-the-art radio sensors and applications connected to the electric grid that can facilitate radiofrequencies for television and the internet to accommodate several technological users. The view toes the line of submissions in [22,27].

## 3. Method and Materials

The following parameters were reviewed in this study as part of the survey's objectives: adaptable areas of Construction 4.0 in infrastructural innovation in design, planning, construction and maintenance; industrial application drivers of I4.0 and C4.0; hindrances in achieving C4.0; achieving the automation dream through C4.0; the social and economic implication of C4.0; the issues and challenges in achieving sustainable innovation in infrastructural development; the disruptive tools of C4.0 in achieving sustainable designs for technological development and the critical factors influencing the effective adaptation of C4.0 in achieving growth. The qualitative research approach was engaged in this study, while random sampling was used to pick the analysis samples. A population frame of 200 construction firms was utilised for the task, while the study included 150 respondents. The population frame consisted of construction firms that are 200 in number.

The population sample for the respondents and population frame for the companies in Abuja and Lagos States in Nigeria were derived using Cochran's formula to arrive at an average of two hundred (200) for the construction firms, with a sample size of one hundred and fifty (150) (obtained from the Corporate Affairs Commission) for the companies sampled, and the formula is as stated below:

$$n_0 = Z^2pq/e^2 \tag{1}$$

where e is the desired level of precision (i.e., the margin of error = 0.05), p is the (estimated) proportion of the population with the attribute in question, q is 1-p and z, the z-value. A confidence level of 80% was used.

The sample size was calculated using Slovin's formula in Equation (2):

$$n\ (1 + Nb^2) = N \tag{2}$$

The respondents used in the study included builders, architects, quantity surveyors, mechanical Engineers and electrical engineers. The authors adopted the simple percentage, relative index, Pearson Spearman's rank, Mann–Whitney U Test, Kendall's Tau test, Student's t-Test, ANOVA and chi-square tools for data processing. Survey research was used in this study to carry out the survey. The administered questionnaire was designed in a Likert scale of 1–5. Relative Agreement Index was calculated for the Likert scale questions using the formula presented in Equation (3).

Relative Agreement Index for each parameter was calculated using the formula below:

$$R.A.I = 5(SA + 4A + 3SD + 2D + 1N)/5(SA + A + SD + D + N) \tag{3}$$

where R.A.I = Relative Agreement Index, SA = Strongly Agree, A = Agree, SD = Strongly Disagree, D = Disagree and N = Neutral.

The breakdown of the objectives with the corresponding analytical methods suitable for each goal is presented below. The following goals/objectives were set as a gap observed in the literature items explored.

Objective 1: To identify the areas of Construction 4.0 (C4.0) that can be adopted to improve infrastructural innovation in design, planning, construction and maintenance. The suitable analytical tools are the Relative Index, Mean Index and Pearson's Spearman's Rank.

Objective 2: To study the industrial application drivers of I4.0 and C4.0 that are important in achieving C4.0 from a professional's perspective. Suitable analytical tools are

Relative Index, Mean Index, Pearson's Spearman's Rank, ANOVA, and Mann–Whitney *U* Test.

Objective 3: Identify strategies that could be used to achieve the inclusive industrial automation development dream through C4.0. Analytical tools for processing the data include Relative Index, Mean Index, Pearson's Spearman's Rank, Chi-square and Friedman's ANOVA.

Objective 4: To study the social and economic implications of C4.0 innovations in industrial development. Analytical tools for the objective are Relative Index, Mean Index and Pearson's Spearman's Rank.

Objective 5: To identify the issues and challenges involved in achieving sustainable design in infrastructural development using C4.0. The analytical tools for the objective include Relative Index, Mean Index and Pearson's Spearman's Rank.

Objective 6: To study the disruptive tools of C4.0 that can be used to achieve inclusive and sustainable innovation for technological development. Analytical tools: Relative Index, Mean Index and Pearson's Spearman's Rank.

Objective 7: To profile the critical factors that influence the practical adaptation of C4.0 in achieving industrial development. The data analysis method includes Relative Index, Mean Index, Pearson's Spearman's Rank, ANOVA and the Mann–Whitney U Test.

Objective 8: To present a pathway for Construction 4.0 and Industry 4.0 for sustainable innovation and inclusive technological development. The analytical tools used for the objective include Relative Index, Mean Index, Pearson's Spearman's Rank and the Mann–Whitney *U* Test.

There are four (4) hypotheses articulated in this study; the hypotheses were drawn from the objectives to corroborate the objectives. The hypotheses were operationalized to illustrate the workability of the objectives. Therefore, the hypotheses were drawn in the following order: Hypotheses 1 and 2 were drawn from objective 1, and Hypothesis 3 was drawn from objective 7, while Hypothesis 4 was drawn from objective 6.

## 4. Results and Presentation

The breakdown of the respondents' Bio data information is as presented in Table 1. The respondents' detailed information covering specific parameters such as the age of the respondents, qualifications of the respondents, designation of the respondents and construction experience are presented. The respondents span between 20 years old and 60 years old. It was discovered that the main bulk of the respondents fall into the 31–40 years old category. Eighty-one (81) respondents, which constituted 54% of the total respondents, were 31–40 years old. Forty-one (41) respondents, which included 27.33% of respondents, belonged to the age range of 20–30 years old. Twenty-one respondents, which constituted 14% of the total respondents, belonged to the age range of 41–50. In comparison, 4.67% of respondents indicated an age range between 51 and 60 years old.

The qualifications of the respondents were also illustrated in Table 1. Eighty-one (81) respondents were construction specialists with degree certifications of Bachelor of Science and Technology constituted 54% of the total respondents. Thirty-two (32) respondents with Master's degree certificates or Masters of Technology included 21.33%. In comparison, twenty (20) respondents that constituted 27.33% had a Higher National Diploma. In comparison, sixteen (16) respondents with PhD degrees constituted 10.6%. The implication of the breakdown indicates a high level of enlightenment on the part of the respondents. The consistency and validity of the analytical results reflect the intellectual base of the respondents.

**Table 1.** Respondents' biodata information.

| A. Years of Experience in Construction Work | Frequency | Percentage (%) |
|---|---|---|
| 1–5 years | 15 | 10.00 |
| 6–10 years | 12 | 8.00 |
| 11–15 years | 21 | 14.00 |
| 16–20 years | 45 | 30.00 |
| Over 20 years | 57 | 38.00 |
| Total | 150 | 100.0 |
| B. Designation of Respondent | Frequency | Percentage (%) |
| Builder | 52 | 34.67 |
| Architect | 40 | 26.67 |
| Quantity Surveyor | 24 | 16.00 |
| Mechanical Engineer | 21 | 14.00 |
| Electrical Engineer | 13 | 8.67 |
| Total | 150 | 100.01 |
| C. Qualification of Respondents | Frequency | Percentage (%) |
| HND | 20 | 13.33 |
| BSc./B.Tech | 82 | 54.67 |
| MSc./M.Tech | 32 | 21.33 |
| PhD | 16 | 10.67 |
| Total | 150 | 100.0 |
| D. Age of Respondent | Frequency | Percentage (%) |
| 20–30 | 41 | 27.33 |
| 31–40 | 81 | 54.00 |
| 41–50 | 21 | 14.00 |
| 51–60 | 7 | 4.67 |
| Total | 150 | 100 |

HND: Higher National Diploma; B.Sc.: Bachelor of Science, M.Tech.: Master of Technology.

The categorisation of the respondents included builders, architects, quantity survey-ors, mechanical engineers and electrical engineers. Therefore, the breakdown included fifty-two (52) builders (34.67%), forty (40) architects (26.67%), twenty-four (24) quantity surveyors (16%), twenty-one (21) mechanical engineers (14%) and thirteen (13) electrical engineers (8.67%). The builders and architects formed the respondents' core; they are directly involved in applying the materials, tools and techniques involved in the design, construction, planning and post-occupancy measurements. These facts further illustrate the validity of the data obtained and presented in this study.

The work experience of the respondents is illustrated in Table 1. The year ranges used in the survey included 1–5 years, 6–10 years, 11–15 years, 16–20 years and over 20 years. Forty-one (41) respondents had 1–5 years' work experience, fifty-seven (57) respondents (38%) had over 20 years of work experience, forty-five (45) respondents (30%) had 16–20 years of experience and twenty-one (21) respondents (14%) had 11–15 years of work experience. In comparison, twelve (12) respondents (8%) had to 6–10 years and fifteen (15) respondents (10) had 1–5 years of construction experience. The data pattern indicates that the highest percentage of respondents had over twenty (20) years of work experience and 11–15 years of work experience, respectively. This gave out reliable information rooted in the respondents' rich work experience.

### 4.1. Identify the Adaptable Areas of Construction 4.0 (C4.0) in Infrastructural Innovation in Design, Planning, Construction and Maintenance

The Adaptable areas of construction 4.0 (C4.0) in Infrastructural Innovation in Design is illustrated in Table 2. The value of the Mean index and Relative agreement index ranking the parameters from highest to lowest in descending order. The adaptable areas in C4.0 as censored in this survey is as shown with corresponding Mean index values. The areas identified are ordered as follows from the highest to lowest points. It includes: Planning,

construction and maintenance telematics equipment and tools with a Mean Index value of 4.655 and ranked 1st, GPS Positioning equipment with Mean index of 4.485 was ranked 2nd, Biomimetic design models with Mean Index 4.435 and Virtual reality software and applications with 4.435 were ranked third, respectively.

**Table 2.** Adaptable Construction 4.0 (C4.0) in Infrastructural Innovation in Design, Planning, Construction and Maintenance.

| Adaptable Areas | Relative Agreement Index | Mean Index | Ranking |
|---|---|---|---|
| Telematics equipment and tools | 0.931 | 4.655 | 1st |
| GPS positioning equipment | 0.897 | 4.485 | 2nd |
| Biomimetic design models | 0.887 | 4.435 | 3rd |
| Virtual reality software and applications | 0.887 | 4.435 | 3rd |
| Artificial Intelligence simulation tools | 0.873 | 4.365 | 5th |
| Parametric modelling | 0.773 | 3.865 | 6th |
| Point cloud technology | 0.773 | 3.865 | 6th |
| Augmented reality tools | 0.771 | 3.855 | 8th |
| Mobile construction education | 0.679 | 3.395 | 9th |
| Digital planning and design applications | 0.677 | 3.385 | 10th |
| Digital costing applications | 0.567 | 2.835 | 11th |
| Knowledge and innovation transfer applications | 0.531 | 2.655 | 12th |

Artificial Intelligence Simulation Tools with Mean Index score of 4.365 were ranked fifth. Parametric modelling with Mean Index 3.865 was ranked 6th with Point cloud technology with a Mean Index value of 3.865 6th. Augmented reality tools of Mean Index score of 3.855 was ranked eighth. Mobile construction education was ranked 9th with Mean Index score of 3.395. Digital planning and design applications were ranked 10th with Mean Index of 3.385, digital costing applications of Mean Index 2.835 were rated 11th. In contrast, Knowledge and innovation transfer applications were ranked 12th with a Mean Index score of 2.655.

*4.2. Investigates Industrial Application Drivers of I4.0 and C4.0 That Are Important in Achieving C4.0 from Professionals Perspective*

Industrial Application Drivers of I4.0 and C4.0 and Hindrances in Achieving C4.0 was presented. Table 3 indicates the industrial application driver variables' ranking to the mean index and relative index from professionals' perspective. automated design system with an average mean index of 0.851 was ranked first. E-procurement system was ranked 2nd with Average Mean Index scores of 0.825. Electronic monitoring with mean index scores of 0.792 were ranked third. Additionally, E-planning applications with Average Mean index of 0.715 were ranked fourth. E-costing using cost software Average mean index value of 0.689 was ranked fifth. E-maintenance using maintenance software with Average Mean Index 0.666 was ranked sixth. Post-occupation management Application with Average Mean Index score of 0.610 and was ranked seventh. The data spread above illustrates the integrity and importance attached to drivers that influence I4.0 and C4.0. The automatic design system was the most preferred among the drivers. Hindrance Parameters were ranked by the three categories of respondents using the respondents' average mean, i.e., the Architects, Builders, Quantity surveyors and Engineers. Government policy is one of the variables. The Average Mean Index of 0.881 was ranked 1st alongside substandard applications with Average Mean Index 0.831st. Fund Scarcity with Average Mean Index of 0.806 was ranked 3rd, while the cultural factor with the mean value of 0.780 was ranked fourth. The ranking of Government policy as first among the hindering forces could be linked to the current political situation as obtainable at the study location, Nigeria, where the data was collated.

There has been policy limitation on the extent of foreign technological component, thereby encouraging local content initiative.

**Table 3.** Industrial Application Drivers of I4.0 and C4.0 that are important in Achieving C4.0 from Professionals' Perspectives.

| Industrial Application Driver | Architect | | Builder | | Quantity Surveyor | | Engineer | |
|---|---|---|---|---|---|---|---|---|
| | Mean | Rank | Mean | Rank | Mean | Rank | Mean | Rank |
| Automated design system | 0.893 | 1st | 0.883 | 1st | 0.756 | 1st | 0.873 | 1st |
| E-procurement system | 0.877 | 2nd | 0.813 | 3rd | 0.773 | 2nd | 0.833 | 2nd |
| Electronic monitoring | 0.873 | 3rd | 0.823 | 2nd | 0.657 | 3rd | 0.812 | 3rd |
| E-planning applications | 0.768 | 4th | 0.765 | 4th | 0.593 | 4th | 0.735 | 4th |
| E-costing using cost software | 0.678 | 5th | 0.763 | 5th | 0.553 | 5th | 0.763 | 5th |
| E-maintenance using maintenance software | 0.677 | 6th | 0.731 | 6th | 0.524 | 6th | 0.732 | 6th |
| Post-occupation-management application | 0.579 | 7th | 0.678 | 7th | 0.521 | 7th | 0.661 | 7th |
| Government policy | 0.888 | 1st | 0.875 | 1st | 0.887 | 1st | 0.875 | 1st |
| Substandard applications | 0.873 | 2nd | 0.765 | 2nd | 0.823 | 2nd | 0.875 | 1st |
| Fund scarcity | 0.773 | 3rd | 0.753 | 3rd | 0.823 | 2nd | 0.873 | 3rd |
| Cultural factor | 0.751 | 4th | 0.731 | 4th | 0.765 | 4th | 0.871 | 4th |
| Social inclusion factor | 0.657 | 5th | 0.695 | 5th | 0.765 | 4th | 0.871 | 4th |
| Gender bias applications | 0.666 | 6th | 0.573 | 6th | 0.731 | 6th | 0.775 | 6th |
| Internet connectivity | 0.561 | 8th | 0.553 | 7th | 0.674 | 7th | 0.773 | 7th |
| Anti-technology transfer policy | 0.563 | 7th | 0.523 | 8th | 0.621 | 8th | 0.771 | 8th |
| Economic and Social policy | 0.553 | 9th | 0.513 | 9th | 0.601 | 9th | 0.753 | 9th |

The Social inclusion factor with mean index 0.747 was ranked 4th. Gender bias applications with 0.686 were ranked sixth. In contrast, Internet connectivity with a factor 0.640 was ranked 7th, along with the anti-technology transfer policy. The Average Mean Index of 0.620 was ranked 8th for Economics, while the Social system with an Average Mean Index of 0.605 was ranked ninth. The facts presented in the table above are supported in [1,3,7].

### 4.2.1. Industrial Application Drivers of I4.0 and C4.0 That Are Important in Achieving C4.0

Some drivers influence the effectiveness of industry 4.0 and construction 4.0. The drivers provide a measurable timeline that could effectively deliver the output of Industry 4.0 and Construction 4.0. The study subjected one of the research questions captured in one of the objectives to validate further the authenticity of the objective and the relationship among the respondents. The hypothesis is stated below and, also, expressed in Table 4.

**Hypothesis 5.** *There is no Significant Difference in the opinions of Professionals on the rating of Drivers of I4.0 and C4.0.*

**Hypothesis 6.** *There is a Significant Difference in Professionals' opinions on the rating of Drivers of I4.0 and C4.*

The authors carried out the Mann–Whitney U test analysis on the data presented in Table 5. The mean rank spans between 1 and 2. The cross-section of data presented covers the respondents such as Architect, Builder, Quantity surveyor and Engineer. There is a homogenous ranking value among the professionals; it indicates the extent of agreement among the professionals based on their ranking. The breakdown of the Mann–Whitney *U* analysis is further presented in Table 6.

**Table 4.** Analysis of Variance Table (ANOVA) on Industrial Application Drivers of I4.0 and C4.0 is critical in Achieving C4.0.

| Drivers Parameters | | Sum of Squares | Df | Mean Square |
|---|---|---|---|---|
| Auto-design | Between Groups | 0.090 | 6 | 0.015 |
| | Within Groups | 0.000 | 0 | - |
| | Total | 0.090 | 6 | - |
| E-procurement | Between Groups | 0.028 | 6 | 0.067 |
| | Within Groups | 0.000 | 0 | - |
| | Total | 0.029 | 6 | - |
| Electronic Monitoring | Between Groups | 0.027 | 6 | 0.004 |
| | Within Groups | 0.000 | 0 | - |
| | Total | 0.027 | 6 | - |
| E- Planning | Between Groups | 0.028 | 6 | 0.067 |
| | Within Groups | 0.000 | 0 | - |
| | Total | 0.280 | 6 | - |
| E-Costing | Between Groups | 0.067 | 6 | 0.011 |
| | Within Groups | 0.000 | 0 | - |
| | Total | 0.067 | 6 | - |
| E-Maintenance | Between Groups | 0.028 | 6 | 0.067 |
| | Within Groups | 0.000 | 0 | - |
| | Total | 0.280 | 6 | - |
| Post occupation | Between Groups | 0.031 | 6 | 0.005 |
| | Within Groups | 0.000 | 0 | - |
| | Total | 0.031 | 6 | - |
| Government Policy | Between Groups | 0.028 | 6 | 0.067 |
| | Within Groups | 0.000 | 0 | - |
| | Total | 0.0280 | 6 | - |

**Table 5.** Mann–Whitney *U* Test Statistical Parameters on Rating of Industrial Application Drivers of C4.0.

| Statistical | Architect | Builders | Quantity Surveyor | Engineer |
|---|---|---|---|---|
| Mann–Whitney *U* | 0.000 | 0.000 | 0.000 | 0.000 |
| Wilcoxon W | 1.000 | 1.000 | 1.000 | 1.000 |
| Z-value | −1.000 | −1.000 | −1.000 | −1.000 |
| Asymp. Sig. (2-tailed) | 0.317 | 0.317 | 0.317 | 0.317 |
| Exact Sig. 2 (1-tailed Sig.) | 1.000 | 1.000 | 1.000 | 1.000 |

**Table 6.** Pearson's Chi-Square Analysis Table on Hindrances in Achieving C4.0 and I4.0.

| Drivers | Auto-Design | E-Procurement | E-Monitoring | E-Planning | E-Costing | E-Maintenance | Post-occupation | Government Policy |
|---|---|---|---|---|---|---|---|---|
| Chi-square | 0.000 [a] | 0.000 [a] | 0.000 [a] | 0.000 [a] | 0.000 [a] | 0.000 [a] | 0.000 [a] | 0.000 [a] |
| df | 6 | 6 | 6 | 6 | 6 | 6 | 6 | 6 |
| Asymp. sig. | 1.000 | 1.000 | 1.000 | 1.000 | 1.000 | 1.000 | 1.000 | 1.000 |
| Hindrance parameters | Substandard Application | Fund scarcity | Cultural factor | Social inclusion | Gender biases | Internet connectivity | Anti-technology | Economic and social policy |
| Chi-square | 0.000 [b] | 0.000 [b] | 0.000 [b] | 0.000 [b] | 0.000 [b] | 0.000 [b] | 0.000 [b] | 0.000 [b] |
| df | 6 | 6 | 6 | 6 | 6 | 6 | 6 | 6 |
| Asymp. sig. | 1.000 | 1.000 | 1.000 | 1.000 | 1.000 | 1.000 | 1.000 | 1.000 |

Sig: Significance df: Degree of Freedom [a] Grouping Variable: Rank1 [b] They are not corrected for ties.

### 4.2.2. Statistical Parameters on Rating of Industrial Application Driver of C4.0

Statistical parameters for the significance analysis of respondents' opinion related to the rating of application drivers of C4.0. The Asymptotic significance value (2-tailed) for the four types of respondents is 0.317. The Mann–Whitney *U* and Wilcoxon W values are 0.317 and 1.00, respectively. The amounts are more significant than *p*-value 0.05; therefore, the Null hypothesis is rejected; accordingly, there is an agreement in the respondents' ranking order.

Similarly, it revealed that the responses are on the high side of the scale of 1–5. The majority of the respondents belong to those who subscribed to scale scores 4 and 5 rather than the lower ones. The breakdown of statistical results is as presented in Table 5.

### 4.2.3. Pearson's Chi-Square Analysis Table on Hindrances in Achieving C4.0 and I4.0

The implication of respondents rating on the driver's variable and hindrances presented for achieving C4.0 and I4.0 was presented in Tables 6 and 7. As relates to the drivers, the Asymp. Sign" if the "Asymp. Sig." number is less than 0.05, the relationship between the two variables in the data set is statistically significant, but if the number is greater than 0.05, the relationship is not statistically significant then the Null Hypothesis is accepted. Therefore, it could be deduced from the analysis that there is no significant difference in the relationship between the drivers and the perceived ratings of hindrances to the achievement of C4.0 and I4.0.

**Table 7.** Summary of Hypothesis Test on Effectiveness of Distribution of Drivers that are important to I4.0 and C4.0.

| Null Hypothesis | Test | Sig. | Decision |
|---|---|---|---|
| The distributions of auto design | - | - | - |
| E-procurement | - | - | - |
| Electronic monitoring, E-planning | - | - | - |
| E-costing, E-maintenance, | - | - | - |
| Post-occupation, | Related-samples | | |
| Hindrance, | Friedman's | | |
| Government policy | Two-way | 0.000 | Reject the Null Hypothesis |
| Hindrance (Substandard) | Analysis of | | |
| Hindrance (Fund scarcity) | variance by Ranks | | |
| Hindrance (cultural factor) | | | |
| Hindrance (gender bias), | - | - | - |
| Hindrance (internet connectivity), | - | - | - |
| Hindrance (anti-technology) | - | - | - |
| Hindrance (economic and social policy are the same). | - | - | - |
| Hindrance(anti-technology) | - | - | - |
| Hindrance (economic and social policy). | - | - | - |

Asymptotic significance is displayed. The significance level 0.05.

**Hypothesis 7.** *There is a significant difference in the ratings of the Drivers and Hindrances to the achievement of C4.0 and I4.0.*

**Hypothesis 8.** *There is a considerable difference in the ratings of the Drivers and importance to the accomplishment of C4.0 and I4.0.*

As highlighted in the table summary, the results' implication lies in a possible match pattern observed among variables. For instance, the likely hindrances to the most highly rated driver (Automated Design and E-Procurement System) are the Government policy, Sub-standard applications, Fund Scarcity, Cultural factor and Social inclusion factors. The authors also discovered it through the Test of Distribution of the relationships among the variables by carrying out tests such as related samples Friedman's Two-way analysis of variance by ranks. The outcome of the tests indicated that there was an effective distribution among the related variables. Therefore, rejecting the Null hypothesis was imperative on account of the statistical result that was generated.

The authors of [7] alluded that substandard application can impact the mechanical design's effectiveness, diffusion of substandard software applications into the field can bring about low facility performance and waste of money time. Similarly, [8,9] supported the opinion in [7] that similar factors such as Fund Scarcity, Cultural factor and Social inclusion factor influence the effectiveness of deployment of C.40 and I 4.0.

*4.3. Identification of Strategies to Achieving Inclusive Industrial Automation Development Dream through C4.0*

The identification of strategies to achieving inclusive industrial automation development dream through C4.0 are expressed in Table 8. Some of the strategies identified in the study are: Application of Internet of things, application of physical, cyber control systems, introduction of business information modelling [IBM], cloud computing, application of physical cyber control system, application of 4D and 5D in design and construction, application of artificial intelligence and informatics, the introduction of additive manufacturing, rolling out of virtual reality and Knowledge augmentation. The internet of things application was ranked 1st among other factors listed, i.e., highest with mean index value of 4.475, application of physical and cyber control systems with a mean index score of 4.465 was ranked 2nd, Introduction of BIM with mean index 4.450 was ranked by respondents as 3rd and Cloud computing was ranked 4th with a mean index score of 4.415. In contrast, application of 4D and 5D in design and construction was ranked with a mean index value of 4.375. The internet of things was regarded as paramount in attaining technological development inclusive in the construction industry.

**Table 8.** Strategies for Achieving the Inclusive Industrial Automation Development Dream through C4.0.

| Strategy | Mean | Relative Agreement Index | Rank |
|---|---|---|---|
| Internet of things | 4.475 | 0.895 | 1st |
| Application of physical cyber control systems | 4.465 | 0.893 | 2nd |
| Introduction of BIM | 4.450 | 0.890 | 3rd |
| Cloud computing | 4.415 | 0.883 | 4th |
| Application of 4D and 5D in design and construction | 4.375 | 0.875 | 5th |
| Application of artificial intelligence and informatics | 4.265 | 0.853 | 6th |
| Introduction of additive manufacturing | 4.050 | 0.810 | 7th |
| Rolling out of virtual reality, knowledge augmentation | 3.820 | 0.764 | 8th |

According to [12,13], Industry 4.0 has led to the automation of different construction work aspects from the design and planning stage to facility running stage. This development has impacted other part of the construction industry across all essential elements. Similarly, [9,14] opined that BIM application in solving the digitalisation of construction activities' components has brought up tremendous gains in enhanced construction productivity. The studies further submitted that the advent of 3D and 4D digital design and simulation systems has led to an inclusive application of the software with universal application in building design and simulations.

Cloud computing and application of artificial intelligence application packages was another area of C4.0 that was brought about by I4.0. In [10–12,15], it was deduced from the summary of various submissions that cloud computing and artificial intelligence has made a tremendous impact that cut across planning, costing, time measurement variable prediction and other numerous gains. Therefore, to sustain the gains of inclusive application of the features, more attention is needed in knowledge management, knowledge augmentation, technological innovation, diffusion and management. This view was supported in [4,6,11].

### 4.4. Investigate Social Economic Implication of C4.0 Innovations in Industrial Development Proposed Analytical Tools

The socioeconomic implications of construction 4.0 innovations in industrial development were studied and analysed. Cross-referencing and cross-validation of respondents' responses were collated and expressed in Table 9. Construction 4.0 has influenced the ways things are conducted in the construction industry; it was a consensus among researchers that I4.0 has strongly influenced C4.0 in achieving technological development. The influence of C4.0 was censored and researched appropriately, and a symbiotic association was noticed between C4.0 and I4.0. Areas of socioeconomic adaptation of C4.0 on technological development was explored and presented in the table. The highest-rated factor based on respondent consensus agreement is effective diffusion of BIM and systems with a mean score value of 4.825, ranked 1st. Next to be rated was the Paradigm shift with a mean score of 4.485 and was ranked 2nd. Additionally, allowance for a multidisciplinary second approach with a mean score of 4.375 was rated 3rd, while encouraging multilevel interaction in the industry with mean value 4.375 was also ranked 3rd. Additionally, enhancing industrial productivity thirds was ranked 4th with a mean score of 4.365. The creation of a sustainable construction system with mean score 4.355 occupies the 5th position. In contrast, knowledge and skill transfer with a mean score value of 4.315 were ranked 6th, and Increased Human and Nations' GDP with mean value 3.875 were ranked 6th and 7th, respectively.

In advocacy, created by researchers related to contributions in the construction field, every people-oriented development should have socioeconomic implications that are beneficial to masses. For instance, it was posited that every meaningful action should impact people's lives, gender-inclusive, cost-efficient, reliable and results-oriented. Therefore, from the variable presented in the table, effective diffusion of the BIM system, effective diffusion and development. BIM has added value in a practical design system, cost and time prediction and parametric application software evolution to solve fundamental construction problems. In-line with parameters like effective diffusion of BIM and methods, a paradigm shift in the construction process, allowance for a multidisciplinary approach, multilevel interaction in the industry and enhanced industrial productivity as presented in this study which formed significant socioeconomic implications of Construction 4.0 (C4.0) [6,17,18], corroborates the points on a positive note. Similarly, among other things, the following socioeconomics attributes of C4.0 identified such as the creation of sustainable construction system, knowledge/skill transfer, increased human and nations GDP, information, knowledge and people integration, apt integration of resources and technology, effective knowledge management, manmade machine resources integration and the effective management of value chain promotion of gender equality toes the line of submission in [3,6,7,13]. The authors identified a pocket of recommendations in [5,7,13]

strongly advocating the reinforcement of socioeconomic parameters to consolidate further the position of advancement already attained with C4.0.

**Table 9.** Socioeconomic Implications of Construction 4.0 Innovations in Industrial Development.

| Socio-Economic Parameters | Mean | Relative Agreement Index | Rank |
|---|---|---|---|
| Effective diffusion of BIM and systems | 4.825 | 0.965 | 1st |
| A paradigm shift in the construction process | 4.485 | 0.897 | 2nd |
| Allows for a multi-disciplinary approach | 4.375 | 0.875 | 3rd |
| Encourages multilevel interaction in the industry | 4.375 | 0.875 | 3rd |
| Enhanced industrial productivity | 4.365 | 0.873 | 4th |
| Creation of sustainable construction system | 4.355 | 0.871 | 5th |
| Enables knowledge/skill transfer | 4.315 | 0.863 | 6th |
| Increased human and nations GDP | 3.875 | 0.775 | 7th |
| Information, knowledge and people integration | 3.865 | 0.773 | 8th |
| Apt integration of resources and technology | 3.855 | 0.771 | 9th |
| Effective knowledge management | 3.765 | 0.753 | 10th |
| Man-machine-resources integration | 3.765 | 0.753 | 11th |
| Effective management of value chain | 3.715 | 0.743 | 12th |
| Promotion of gender equality | 3.285 | 0.657 | 13th |

*4.5. Examine Issues and Challenges Involved in Achieving Sustainable Innovation in Infrastructural Development*

Investigating and studying issues and challenges involved in achieving sustainable innovation in infrastructural development is presented in Table 10. There are a lot of challenges that are involved in the derivation of benefit derivable from I4.0 and C4.0. The issues are pertinent to the effective deployment and adoption of I4.0 and C4.0. In [5,7], hindrances are described as a bottleneck and pitfalls that should be cleared for an effectively deployed innovation. The authors of [8] illustrates the issues and challenges as dissatisfiers that influence innovation diffusion. Problems that are typical of controlling digital expedition are grouped and regrouped into seven (7) main points, as presented in Table 10. Some of the cases were studied and calibrated using their mean value and Relative Agreement Index (RAI). The psychological attachment was highly rated above others as reflected in Peoples' psychological attachment to old ways of carrying out construction operations. The factor was ranked by the respondents as 1st with a mean index value of 4.45. Other factors included Educational underdevelopment with mean index 3.930 and ranked second. Unwillingness to transfer the skill to learners on projects with a mean value of 3.655 was ranked 3rd, and Government policy with a mean index value of 3.655 was ranked fourth.

Similarly, the unwillingness of construction practitioners to learn new technology with mean score 3.265 was ranked fifth. In contrast, Non-E-readiness of the construction industry with mean index 3.160 and Social and Cultural affinity with mean score 3.155 were organised by the respondents as sixth and seventh. One of the significant resistances to change is psychological attachment. It is regarded as the primary factor. It is always difficult for people to adapt to a new change in a system; this could be traced to educational awareness; the trend follows the submissions as posited in [3,5].

In [17,18], educational underdevelopment was described as a bedrock factor and significant technological development issue. Education is regarded as a critical ingredient in training and technical development all over the world. The more a system is open to educational enlightenment, the more civilised the system and vice versa. Unwillingness to shift learning position is also an issue in technological development; it could be linked

to ignorance or lack of good innovation. Non-E-readiness of construction industry was regarded as another issue; readiness for a change always precedes the acceptance of a design. This is one of the many reasons for digital transformation necessity in a system. It toes the line of submission as presented in [3,5,7,13].

**Table 10.** Issues and Challenges Involved in Achieving Sustainable Innovation in Infrastructural Development.

| Issues and Challenges Issues | Mean | Relative Agreement Index | Rank |
|---|---|---|---|
| People psychological attachment to old ways of doing things | 4.445 | 0.889 | 1st |
| Educational underdevelopment | 3.930 | 0.786 | 2nd |
| Unwillingness to transfer the skill to learners on projects | 3.655 | 0.731 | 3rd |
| Government policy | 3.655 | 0.731 | 4th |
| The refusal of construction practitioners to learn new technology | 3.265 | 0.653 | 5th |
| Non-E-readiness of the construction industry | 3.160 | 0.632 | 6th |
| Social and cultural affinity | 3.155 | 0.631 | 7th |
| Challenges | - | - | - |
| Challenges of digital divide | 4.375 | 0.875 | 1st |
| Challenges associated with cybernetics' | 3.825 | 0.765 | 2nd |
| Challenges of fluctuating power supply | 3.765 | 0.753 | 3rd |
| Dynamics of hackers and cyber fraud | 3.655 | 0.731 | 4th |
| Regional and continental political and economic challenges | 3.475 | 0.695 | 5th |
| Limitations associated with the internet of things | 2.865 | 0.573 | 6th |

Similarly, the challenges that militate against achieving sustainable infrastructural development was illustrated in Table 10. The digital divide was the main challenge identified and ranked 1st with a mean score rating of 4.375. Challenges associated with cybernetics with a mean score of 3.825 were ranked second. In contrast, the fluctuating power supply challenges ranked 3rd with a mean index value of 3.765. Additionally, as a factor, dynamics of hackers and cyber fraud was ranked 4th with mean score 3.655, regional and continental political and economic challenges were ranked 5th and 6th with 3.475. In contrast, limitations associated with the internet of things was ranked 7th with a mean value of 2.865.

In line with [1,5,11,12], digital divide and cybernetics challenge are among the biggest challenges that need utmost attention. There is still a restriction in applying digital technology from one continent to another, based on its policy in such a continent. Similarly, cyber regulation and cyber security are of the essence in maintaining serenity in digital technology application.

*4.6. Disruptive Tools of C4.0 in Achieving Inclusive Sustainable Innovation for Technological Development*

Disruptive tools of C4.0 in achieving inclusive, sustainable innovation for technological development are presented in Table 11 as a data spread of disruptive tools. The tools include design tools, construction tools, sensor-based, security system and artificial intelligence manufacturing tools, according to the data spread in the table generated through the excellent experience of construction professional who has used the application that

contains the tools, spread through a wide range of applications. The essential tools rated is Sensor-Based Hand Tools that was ranked first unanimously by the three categories of the respondents. The sensor-based tools are predominantly used in the quality assurance system during product manufacturing system. Most of the Toyota production system's quality assurance process and most of the Job flow process in an industrial production system often involves extensive product manufacturing process [24–26].

**Table 11.** Disruptive Tools of C4.0 in Achieving Inclusive Sustainable Innovation for Technological Development.

| Disruptive Tools of C4.0 | Builders | | Architect | | Quantity Surveyor | |
|---|---|---|---|---|---|---|
| | Mean | Rank | Mean | Rank | Mean | Rank |
| Sensor-based hand tools | 0.893 | 1st | 0.893 | 1st | 0.887 | 1st |
| Blended technology | 0.892 | 2nd | 0.877 | 2nd | 0.879 | 2nd |
| Blended application tools | 0.891 | 3rd | 0.873 | 3rd | 0.873 | 3rd |
| Telemetric applications | 0.887 | 4th | 0.768 | 4th | 0.757 | 4th |
| Flipped technology | 0.883 | 5th | 0.678 | 65h | 0.661 | 5th |
| Radio sensor equipped security system | 0.875 | 6th | 0.677 | 6th | 0.632 | 6th |
| Digital hammer | 0.853 | 7th | 0.579 | 7th | 0.623 | 7th |
| Artificial intelligence tools | 0.764 | 8th | 0.569 | 8th | 0.573 | 8th |

In [20,23], the importance of sensor-based tool was highlighted to automate necessary traditional tools to an automated type through improved internal efficiency for an enhanced product manufacturing. Blended technology was ranked 2nd, while the respondents third-ranked blended Telemetric applications. Blended application is adapted in construction knowledge learning and impartation. Construction education is being carried out conventionally now with the aid of Blended learning tools. Blended application tools involves adopting tools like Smart boards, Smart screens, podcast screens, I-Pods, I-casts and virtual reality gadgets, among others. The application allows for the interchange of data and information real-time and offline. It assists in data communication, visualisation and presentation [27–30].

Additionally, the Telematics tool was ranked fourth. Telematics combines the instrumentality of construction informatics with telecommunications. Telecommunication system and information system are fused into a typical Telematics software loaded on machines and an articulated plant and equipment on sites. The view is supported in [8–10]. Flipped Technology assists technician and construction professionals to communicate effectively on-site. According to [9,10,13], mobile applications operate flipped technology that enables dual communication of receiving and calling. The technology has enabled mobile technology modules that can forward pictures and graphic virtually on site. Other subsidiary tools in the application include radio sensor equipped security system, digital hammer and artificial intelligence tools. The findings toe the line of submissions in [3,7,11].

The relationship was further explored using Wilcoxon and Mann–Whitney *U* statistical tools. In contrast, the statistical results are presented in Table 12.

**Table 12.** Wilcoxon and Mann–Whitney *U* statistical test results.

| Statistical Parameters/Ch-aracteristics | Sensor-Based Hand Tools | Blended Technology | Blended Application Tools | Telemetric Applications | Flipped Technology | Radio Sensor-Equipped Security System | Digital Hammer | Artificial Intelligence Tools |
|---|---|---|---|---|---|---|---|---|
| Mann–Whitney *U* | 0.000 | 0.000 | 0.000 | 0.000 | 0.000 | 0.000 | 0.000 | 0.000 |
| Wilcoxon W | 1.000 | 1.000 | 1.000 | 1.000 | 1.000 | 1.000 | 1.000 | 1.000 |
| Z-value | −1.000 | −1.000 | −1.000 | −1.000 | −1.000 | −1.000 | −1.000 | −1.000 |
| Asymp.Sig. (2-tailed) | 0.317 | 0.317 | 0.317 | 0.317 | 0.317 | 0.317 | 0.317 | 0.317 |

**Hypothesis 9.** *There is no uniformity of opinions on Disruptive tools of C4.0 Adaptation in Achieving Sustainable Technological Development.*

**Hypothesis 10.** *There is Uniformity of Opinions as regards Disruptive mechanisms of C4.0 Adaptation in Achieving Sustainable Technological Development.*

In the data analysis presented in Table 12, the respondents' cosmopolitan nature was considered and explored for the universality of thought and agreement on the issue addressed in the table. Wilcoxon W results indicated 1.000 universal value, while 0.000 was obtained in the case of the Mann–Whitney *U* analysis. However, the Asymptotic significance (2-tailed) was of a maximum weight of 0.317. In any case, the Asymp. Sig. was more significant than the *p*-value 0.050; thus, the Null hypothesis should be aborted. Therefore, it was inferred that there are standard agreement and universality of thought and opinion among the respondents. This, according to [6,9,11], they expressed the reason behind this as a reflection a pattern like this whenever they occur as being linked to universal experience garnered on-site because of time. The design obtained illustrate the reason behind the following parameter ratings: Sensor-Based Hand Tools, Blended Technology, Blended Application Tools a Telemetric Applications.

*4.7. Profile of Critical Factors Influencing Effective Adaptation of C4.0 in Achieving Development*

Construction 4.0 is an exciting concept that has influenced the construction arena seriously. The impact of C4.0 has led to tremendous improvement in design, planning, construction and maintenance of building and facilities. However, for a holistic and effective adaptation of structure 4.0 for proper positioning to achieve industrial development, there are pertinent factors that need full consideration and essential to achieving industrial action as posited in the submissions of [7,8,11,14]. The three authors support the opinion that hinges meaningful infrastructural development and integration for effective delivery of C4.0. However, [11,13] argued that policy formulation is required for an impactful industrial development expected through C4.0 adaptations. It has to be a holistic one that cut across various concepts and factors like the documented types and is presented in Table 13. Some of the significant identified factors include the Evolution of corporate identity with an average mean score of 4.480 and ranked 1st by the three types of respondents used in this study.

Construction 4.0 should be accorded an identity by users, which should be synchronised and reflect the kinds of value being created and identity created for it by the relevant stakeholders. Similarly, the Vertical integration of services and products and the possibility of technology and skill transfer were ranked second and 3rd, respectively, with average mean scores of 4.428 and 3.976 in a similar order.

Similarly, the growth of Small scale enterprises (SME) should be supported. SME is regarded as a foot soldier whose support advertises the gains of C4.0. Therefore, keeping their development is essential. It is highly imperative for technological advancement and diffusion of SME. In the works of [22,23], SME was regarded as the vehicle of adoption, testing, and innovation integration meant for diffusion into the construction industry.

**Table 13.** Factors Influencing Effective Practical Adaptation of Construction 4.0 in Industrial Development.

| Influencing Factors | Architect | | Builders | | Quantity Surveyor | | Engineer | |
|---|---|---|---|---|---|---|---|---|
| | Mean | Rank | Mean | Rank | Mean | Rank | Mean | Rank |
| Evolution of corporate identity | 4.480 | 1st | 4.480 | 1st | 4.085 | 1st | 4.254 | 4th |
| Vertical integration of services and products | 4.435 | 2nd | 4.450 | 2nd | 3.785 | 2nd | 4.444 | 1st |
| Possibility of technology and skill transfer | 4.430 | 3rd | 3.915 | 6th | 3.925 | 3rd | 3.630 | 4th |
| Support for Small scale enterprizes growth | 4.425 | 4th | 3.875 | 7th | 3.765 | 4th | 3.855 | 3rd |
| Enhanced industrial productivity | 4.365 | 5th | 4.325 | 3rd | 3.815 | 5th | 3.375 | 6th |
| Encouragement of stakeholders competition | 4.350 | 6th | 4.405 | 5th | 3.770 | 6th | 4.405 | 2nd |
| Rapid industrial and corporate growth | 4.285 | 7th | 4.175 | 4th | 3.789 | 7th | 3.325 | 7th |
| Advancement in knowledge and technology | 4.265 | 8th | 3.765 | 8th | 3.655 | 8th | 2.890 | 8th |

Additionally, enhanced industrial productivity favouring C4.0 adoption and application is essential while creating a system that encourages stakeholders' competition is very necessary, as opined in [7,13,17]. The authors of [22] favour stimulating rapid industrial and corporate growth. Simultaneously, [23–25,30] supports advancements in knowledge and technology as a panacea to significant, result-oriented technology development and application.

*4.8. The Rating of Effective Adaptations of C4.0 in Technological Development*

As relates to the drivers presented in Table 14 above, the Asymptote Significance (Asymp. Sig.). If the "Asymp. Sig." number is less than 0.05, the relationship between the two variables in the data set is statistically significant. However, if the number is greater than 0.05, the relationship is not statistically significant; then, the Null hypothesis is accepted. Therefore, it could be deduced from the analysis that there is no significant difference in the rating of Effective adaptations of C4.0 in Technological development. The implication of this lies in a substantial relationship pattern that was observed to exist among the variables [31–33]. Hypothesis testing of effective practical adaptations of C4.0 in technological development was presented in Table 14. The results of analysis indicated the Mann Whitney U statistical value to be 0.000 while the Wilcoxon statistics was 1.00 while the Z-value indicated a degree of statistical stability.

**Table 14.** Rating of Effective Adaptations of C4.0 in Technological Development.

| Statistical Parameters | Architect | Builders | Quantity Surveyor | Engineer |
|---|---|---|---|---|
| Mann–Whitney *U* | 0.000 | 0.000 | 0.000 | 0.000 |
| Wilcoxon W | 1.000 | 1.000 | 1.000 | 1.000 |
| Z-value | −1.000 | −1.000 | −1.000 | −1.000 |
| Asymp. Sig. (2-tailed) | 0.317 | 0.317 | 0.317 | 0.317 |
| Exact Sig. [2-tailed Sig.] | 1.000 [b] | 1.000 [b] | 1.000 [b] | 1.000 [b] |

[b] Not corrected for ties.

**Hypothesis 11.** *There is no agreement on the ratings of Effective Practical Adaptations of C4.0 in Technological Development.*

**Hypothesis 12.** *There is agreement on the ratings of Effective Practical Adaptations of C4.0 in Technological Development.*

*4.9. Importance of Construction 4.0 (C4.0) in Achieving Inclusive Technological Development Planning, Construction and Maintenance*

The importance of Construction 4.0 (C4.0) in achieving an inclusive technological development is expressed in Table 15. The content of the table indicates the aspect where C4.0 could be beneficial in attaining inclusive technological development. Some of the areas covered areas presented in the table; they include communication, industrial action, skill and innovation transfer and project success. C4.0 tools have potential to enhance the communication effectiveness in the construction industry. For instance, Telescopic appliances are installed on articulated plant and machines for better output. Telemetric and night vision radar control devices enable good vision at night on some types of Bulldozers and included plants and machines. Similarly, 3D and 4D-enabled appliances that are being engaged at the design and planning stages change the game plan of designs at the design, planning, construction and post-occupation building location. Therefore, the tools of C4.0 can enhance the exchange of better construction information and ideas on construction projects. This could account for the factor being rated as 1st with a mean index value of 4.480 and toes the line of submission in [3,5,9,15], which further stressed the importance of industrial 4.0 and C4.0 applications [31–33].

**Table 15.** Importance of Construction 4.0 (C4.0) in Achieving Inclusive Technological Development Planning.

| Importance of Construction 4.0 in Inclusive Technology Development | Mean | Relative Agreement Index | Rank |
|---|---|---|---|
| It would lead to an exchange of better construction information and ideas. | 4.480 | 0.897 | 1st |
| It would lead to Industrial Development. | 4.436 | 0.887 | 2nd |
| Construction 4.0(C4.0) would enable timely completion of a project. | 4.411 | 0.882 | 3rd |
| There is a tendency for technological skill transfer through C4.0 practice. | 4.306 | 0.861 | 4th |
| Construction 4.0(C4.0) is the panacea to cost overrun on construction projects. | 4.123 | 0.825 | 5th |

There is an agreement in the part of respondents that C4.0 would lead to Industrial Development. Thus, the factors were rated 2nd with mean index scores of 4.436 and closely related to this factor. This factor states that Construction 4.0 (C4.0) would enable the timely completion of a project and is rated 3rd with a mean index of 4.411. The authors of [13,14] mentioned in their submission the advent of a plethora of apt and can help planners overcome time and cost overrun on construction projects. BIM application, one of the C4.0 products, has helped in software development and enhanced cost preparation on sites. Similarly, there is a tendency for technological skill transfer through C4.0 practice. The authors of [7,15,23,34] posited on tools of I4.0 that have helped achieve C4.0; the technology transferred through I4.0 has led to functional adaptation that led to I4.0 tools being engaged to bring about Construction 4.0 application. Ideally, there is always a tendency for skill transfer on projects where the participants can benefit in the direct application of such technology being engaged. There is still the likelihood of intentional skill transfer when the technology and skill champion has decided to allow the transfer. However, restriction

in the scope and limitation of skill meant to be transferred depends on what is contained in the project's memorandum of agreement [35–37].

### 4.10. Pathways of Achieving Sustainable Innovations and Inclusive Technological Development through Construction 4.0 and Industry 4.0

The pathway parameters for achieving sustainable innovations and inclusive technological Development through Construction 4.0 and Industry 4.0 were presented in this study.

The pathway parameters for achieving sustainable innovations and inclusive technological Development through Construction 4.0 and Industry 4.0 are presented in Table 16. The pathway was coined from the Construction 4.0 and Industrial 4.0 parameters. Some of the parameters used are as follow: engaging industrial application drivers, vertical integration of innovation and adoption strategy, engaging standard application of software and system, proactive training of personnel, initiating and adoption of industry 4.0 and construction 4.0, integration of information, knowledge and people, adoption of inclusive industrial application strategies, multilevel interaction of components in the construction industry, enabling knowledge and skill transfer, gender inclusiveness in technological development, encouraging sustainable construction system, effective knowledge management and horizontal integration of consultative information.

**Table 16.** Pathways of achieving Sustainable Innovations and Inclusive Technological Development

| Pathway Parameters | Relative Agreement Index | Mean Index | Rank |
|---|---|---|---|
| Engaging mission-oriented innovation application drivers | 0.894 | 4.470 | 1st |
| Vertical Integration of Innovation and adoption strategy and circular economy | 0.893 | 4.465 | 2nd |
| Engaging Standard application of software and system | 0.893 | 4.465 | 2nd |
| Proactive training of personnel | 0.891 | 4.455 | 4th |
| Initiating and adoption of Industry 4.0 and Construction 4.0 | *0.891* | 4.455 | 4th |
| Integration of information, knowledge and people | 0.887 | 4.435 | 6th |
| Adoption of inclusive industrial application strategies | 0.885 | 4.425 | 7th |
| Multilevel interaction of components in the construction industry | 0.875 | 4.375 | 8th |
| Enabling knowledge and skill transfer | 0.853 | 4.265 | 9th |
| Gender inclusiveness in technological development | *0.850* | 4.250 | 10th |
| Encouraging sustainable construction system | *0.785* | 3.925 | 11th |
| Effective knowledge management | *0.765* | 3.825 | 12th |
| Horizontal integration of consultative information | 0.750 | 3.750 | 13th |

### 4.11. Pathways of Achieving Sustainable Innovations and Inclusive Technological Development

As part of the advocated pathway, as presented in this study, a collaborative approach is leveraging on the professionals' objective submissions. The first action supported as a way of achieving sustainable innovations and technological development is engaging mission-oriented innovation application drivers, thereby ranked 1st with a mean index 4.470, as reflected in Table 16. There should be a delineation of scope and focus of applying innovation and strategies to be adopted. The type of mission that is to be achieved and

the centre of focus of achievement alongside the audience should be identified. The innovation that would be sustainable and durable must incorporate the actors, drivers and audience/consumer. The consumers and end-users need for a commodity or design is pertinent to a system's effectiveness. It is a significant key to a technology or innovation's shelf life and setting up a plan's mission, thereby ensuring its sustainability. The authors of [38,39] posited that the horizon of mission and innovation focus could be enlarged by including networks and socioeconomic movements. It involves stakeholders such as academia, practitioners, technologists and industry captains for knowledge innovation, creation and adoption.

As presented in Table 16 above, Vertical integration of innovation and adoption strategy and standard engaging application of software and system is necessary for technology management and administration; the factors were ranked 2nd with a mean index of 4.465. In [33,34,37]; the importance of technological development was stressed, digital transformation warrants multilevel interaction of stakeholders for the effective exchange of applications. Similarly, standard software is necessary for engagement of functions, thus creating an opportunity for technological development.

Personnel training for effective engagement of technology and innovation is essential; therefore, personnel and end-users need knowledge application orientation. Personnel training was recommended for innovation and technological diffusion in-line with the submissions in [10], as presented in the innovation and distribution theory by Roger Everet. A study carried out by [14,15,38] leveraging digital transformation with associated technologies was stressed as a key to achieving sustainable innovation. The Industry 4.0 and Construction 4.0 applications suggested in [16,17,32] have been observed as game-changers in the application of the digital system in the construction industry and industrial production and manufacturing. The integration of systems should result in the proactive training of personnel and Initiating and adopting Industry 4.0 and Construction 4.0; the factors were scored with a mean index of 4.455, respectively.

In this study's context, the following factors formed the pathway's contents: Integration of information, knowledge and people, adoption of inclusive industrial application strategies, multilevel interaction of components in the construction industry and enabling experience and skill transfer. Additionally, gender inclusiveness in technological development encourages sustainable construction, effective knowledge management and the horizontal integration of consultative information.

## 5. Discussion

The centre of discussion in the context of this study has revolved around three critical axioms: Disruptive adaptations of Construction 4.0, Industry 4.0 as a pathway to a Sustainable Innovation and Inclusive Industrial Technological Development. Disruptive Adaptations of Construction 4.0 was explored, highlighting important areas that impact infrastructural development. Disruptive applications of I4.0 was engaged in manufacturing of components that are used in industrial application manufacturing. There are product design and calibration systems that leverage on I4.0 applications. For instance, Artificial intelligence has been embedded in some applications that engaged sensor-based tools for effective operations. In Construction 4.0, advanced tools are employed, which has led to tremendous success in applying technological tools toward improving productivity. In recent times, Architects have engaged in ArchiCAD in design, ngineers engage Revit and Orion, which leverage BIM innovations [40–42].

Similarly, Revit has been a tool that helped carry out the simulation process of reality as posited in [15,17,41]. Knowledge augmentation produces voice-to-graphic and graphic-to-voice applications, which have assisted in voice-enabled applications. The inclusiveness of the applications lies in the interoperability of the C4.0 applications' functional components that create a system with its applicability spanning across different facets of a system. This submission draws strength from related submissions, such as in [15,17,30]. Some of the essential tools are documented in [1,9,23], which applications cut across the adaptable

areas of Construction 4.0 in infrastructural innovation in design, planning, construction and maintenance.

Similarly, Industry 4.0 has proven to be a panacea for creating a pathway to a Sustainable Innovation design, development and diffusion/application; in creating a path for an inclusive application that would enable sustainable innovation for infrastructural development, a quantitative and qualitative approach is required. In developed countries in Europe, Asia and America, innovations play an active role in carrying out developmental and infrastructural development activities. There has been the advent of new design methods and processes coming up from the advanced economy. It varies from one continent to the other. For instance, applying Artificial intelligence in construction works has been most prevalent and familiar in America and Germany. Companies are already organising training for workers to spread innovation. In Singapore, an application uses Augmented reality that simulates the behaviours of safety devices, plumbing systems and electrical systems that integrate building components; this is corroborated in the works of [3,4,7,12,13]. However, specific essential criteria need to be considered when creating a formidable pathway for sustainable innovations that could lead to technological development. Some of the requirements include identifying industrial application drivers of I4.0 and C4.0, identifying the hindrances in achieving C4.0, censoring issues and challenges in achieving sustainable innovation in infrastructural development and developing disruptive tools of C4.0 in achieving sustainable design for technological development. In the context of this study, the parameters that influence the pathway creation are listed.

Some parameters drive the creation of the pathway, such as automated design system, e-procurement system, electronic monitoring, e-planning applications, e-costing using cost software, e-maintenance using maintenance software and post-occupation-management application; this view is supported in [1], listing digital technologies applicable to industrial development, and further corroborated in [3,7].

Moreover, the importance of inclusive industrial, technological development was explored and documented. Many factors influence inclusive technological ventures. The elements also help achieve the automation dream through C4.0 while still leveraging the social and economic implications of C4.0 cutting across impact from cities to grass roots. In the context of this study, achieving inclusive industrial technological developments involve leveraging on the following areas of application as submitted in [13,14,23]. The areas are sensor-based hand tools, blended technology, blended application tools, telemetric applications, flipped technology, radio sensor equipped security system, digital hammer and artificial intelligence tools. This submission toes the line of the presentations in [24,28] that encompass the usable digital tools, [31] blended technology and [33] leveraging digital technology. The technological developments that can evolve through the applications above includes consumer applications, manufacturing applications, commercial applications, maintenance applications, design and construction applications and construction education. The socioeconomic implications of inclusive industrial technology as posited in [3,6,17,18] includes the effective diffusion of BIM and systems, a paradigm shift in the construction process, allowance for a multidisciplinary approach and a multilevel approach interaction in the industry, enhancing industrial productivity and the creation of a sustainable construction system, among others.

## 6. Conclusions

Some goals/objectives were set at the beginning of this study and formed the focus of discussion in this section. The areas of Construction 4.0 (C4.0) that can be adopted to improve infrastructural innovations in design, planning, construction and maintenance were discovered. The areas identified included planning, construction and maintenance telematics equipment and tools, GPS positioning equipment, biomimetic design models, virtual reality software and applications, artificial intelligence simulation tools, parametric modelling, point cloud technology, augmented reality tools, mobile construction education,

digital planning and design applications, digital costing applications and knowledge and innovation transfer applications.

The drivers of industry 4.0 and Construction 4.0 and the hindrances in achieving Construction 4.0 were identified and profiled. The main drivers of efficiency identified in this study included Automated Design System, E-procurement system, Electronic Monitoring, E-planning applications and E-costing using cost software. In the study, the strategies used to achieve the inclusive industrial automation development dream through C4.0 were presented. The application of the internet of things was rated high, then the applications of physical and cyber control systems, the introduction of BIM, cloud computing, application of physical, cyber control system, applications of 4D and 5D in design and construction, application of artificial intelligence and informatics, the introduction of additive manufacturing, rolling out of virtual reality and knowledge augmentation, among others.

Regarding the corporate and social contributions of Construction 4.0 (C4.0), the study profiled and presented the social and economic implications of Construction 4.0 innovations in industrial development. Construction 4.0 has economic and social benefits. It enhances economic growth and social cohesion in this study's context, the socioeconomic implications of the adoption and deployment of Construction 4.0. The following areas of contribution of C4.0 to the economic and social components of society, effective diffusion of BIM and systems and a paradigm shift in the construction process allow for a multidisciplinary approach, encourages multilevel interaction in the industry, enhanced industrial productivity and the creation of sustainable construction system and enables knowledge/skill transfer; this toes the line of the submissions in [31,33]. Technology integration into society improves technological applications and leveraging technology for creating an awareness of problem-solving skills to engineering students.

We identified the issues and challenges involved in achieving sustainable designs in infrastructural development using C4.0. [26,33,39]. Part of the issues identified were related to the operations of Construction 4.0, issues like peoples' psychological attachments to the old ways of doing things; educational underdevelopment and unwillingness to transfer the skill to learners on projects, government policy and the refusal of construction practitioners to learn new technology. The authors of [26,39] argued about the peculiarity of the issue as being location-dependent; for instance, [26] commented on Construction 4.0 as being the future of technological development in Germany. Construction 4.0 has changed construction events in Germany and Bavarian regions, since Construction 4.0 appeared in the German construction industry. However, challenges confronting the implementation and deployment of Construction 4.0 technology include challenges of the digital divide, challenges associated with cybernetics, challenges of fluctuating power supplies, dynamics of hackers and cyber fraud, regional and continental political and economic challenges and limitations associated with the internet of things. This is in order with the submissions in the works of [13,15,17], stressing the impacts and restrictions in the adoption and implementation of technologies considering the digital divide.

As part of the objectives set at the outset of this study, finding out about the tools of Construction 4.0 provides an amicable path to fulfilling the goals of Constriction 4.0. Disruptive tools of C4.0 that can be used to achieve inclusive and sustainable innovation for technological development were identified. Some instruments were identified as the primary tools for disruptive activities in society. Sensor-based hand tools were identified as being highly deployed. Many of machines in use are equipped with state-of-the-art sensors that encourage automation in product manufacturing and management. Other tools include blended technology, blended application tools, telemetric applications, flipped technology, radio sensor-equipped security system, digital hammer and artificial intelligence tools. These submissions toe the line of the contributions in [10,16,26,34].

Similarly, some factors influence the practical adaptation of C4.0 in achieving industrial development. The elements were examined and articulated in this study; they are the nuggets that influence success in the application and deployment of Construction 4.0. The factors are listed in the order of importance; they include the evolution of corporate

identity, vertical integration of services and products, possibility of technology and skill transfer, support for SME growth, enhanced industrial productivity, encouragement of stakeholder competition, rapid industrial and corporate development and advancement in knowledge and technology. Some of the factors were listed according to the views expressed in [10,15,16] as relevant for achieving sustainable infrastructures being engineered by Industry 4.0 and Construction 4.0.

Generally, this study presents a Construction 4.0 and Industry 4.0 application pathway for sustainable innovation and inclusive technological development. The path offers a systematic order of what should be monitored while adopting Construction 4.0 and Industrial 4.0; the pathways were summarised as follows: engaging mission-oriented innovation application drivers; vertical integration of innovation and adoption strategies; engaging standard applications of softwares and systems; proactive training of personnel; initiating the adoption of industry 4.0 and construction 4.0; the integration of information, knowledge and people; adoption of inclusive industrial application strategies; multilevel interactions of components in the construction industry; enabling understanding and skill transfers; gender inclusiveness in technological development; encouraging a sustainable construction system; effective knowledge management and the horizontal integration of consultative information. Some studies corroborated the pathway's components presented in [10,16], mentioning skill transfer and knowledge management. Simultaneously, [33,34,37] favoured the proactive training of personnel, gender inclusiveness in technological development and engaging mission-oriented innovation. Industry 4.0, combined with Construction 4.0, has proven to be a solution to the provision of smart cities, technology and infrastructure. It has been proven to hold the keys to the fulfilment of sustainable infrastructure and technological development, as it is currently experienced in developing and developed nations. This study has expressed how a hybrid pathway can help create a protocol adopted to achieve sustainable infrastructure and assist in technological development.

The study has explored the concept of Industry 4.0 and Construction 4.0, which was used to create a pathway that could be used to achieve the UNDP goals 9 and 11, which include the provision of sustainable infrastructure and technological development novel studies.

**Author Contributions:** Conceptualisation, A.L. and J.O.; methodology, A.L.; software, A.L.; validation, A.C. and J.O.; formal analysis, A.L.; investigation, A.L.; resources, A.L.; writing—original draft preparation, A.L.; writing—review and editing, visualisation, J.O.; supervision, A.C.; project administration, A.L. and funding acquisition, A.L. All authors have read and agreed to the published version of the manuscript.

**Funding:** This research was funded by the Center for Research, Innovation and Discovery (CUCRID) Covenant University. Ota. Ogun State Nigeria.

**Institutional Review Board Statement:** Not Applicable.

**Informed Consent Statement:** Informed consent was obtained from all subjects involved in the study.

**Data Availability Statement:** The study did not consult supporting data for the study.

**Acknowledgments:** The authors express appreciation to the Center for Research, Innovation and Discovery (CUCRID) Covenant University. Ota. Ogun State Nigeria for funding the Open Access fee and APC for this publication. Additionally, they express appreciation to the Building Informatics Research Sub-cluster of Building Technology Department, Covenant University, which supported this work's success.

**Conflicts of Interest:** The authors declare no conflict of interest. The funders had no role in the study's design; in the collection, analyses or interpretation of data; in the writing of the manuscript or in the decision to publish the results.

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
