# Peer review of "The Disruptive Adaptations of Construction 4.0 and Industry 4.0 as a Pathway to a Sustainable Innovation and Inclusive Industrial Technological Development"

_buildings, doi:10.3390/buildings11030079_

Round 1

Reviewer 1 Report

The authors deal with an interesting topic of Construction 4.0 as a pathway to achieve sustainable innovations and inclusive technological and infrastructural development as a part of Industry 4.0. Although the topic is very interesting, there are several major insufficiencies that need to be improved. These insufficiencies can be summed up in poor scientific writing.

Suggestions for improvement:

  • Check and improve the English language and grammar throughout the paper (check spelling mistakes, writing in the first person, etc.) as well as all figures and tables (both must be readable), define abbreviations before the first use, do not start the section with figure or table, etc.
  • The abstract is a bit ambiguous. Authors are advised to be concise and to keep it short. Also, highlight the goals and findings
  • The introduction section does not provide sufficient background and includes all relevant references. The used references are not novel and some fundamental references are missing (line 34 and 70), as well as the recent ones that take the research problem into account. Are these studies important one or are there any more that are connected to this particular research problem? Also, is there a difference between “development goal 9 and 11” and “objectives 9 and 11”? Please clarify! In lines 94 – 116 it is not clear the need for highlighting “proposed analytical tools” so many times. The research problem is clear, but research goals and hypotheses are not clearly stated
  • Line 117: the title is a bit odd and unusual
  • The methods and materials section is a bit short. Please consider highlighting what is the sole purpose of addressed methods and how they will help you to achieve research goals and hypotheses
  • The results section is the strongest one. There are some misspelling and minor technical problems (see the first bullet)
  • At the moment there are some good observations and arguments in the discussion section. The authors are urged to draw conclusions that are more specific. There should be a clear connection with the research problem, goals, and results
  • The paper should be set according to the Journal’s template and instruction to authors (text, figures, tables, references, etc.). Authors are advised to be consistent throughout the manuscript

Overall, I strongly urge the author to reconsider the above comments, rewrite the paper accordingly, and resubmit it.

Reviewer 2 Report

Review Report

Manuscript

Disruptive Adaptations of Construction 4.0 and Industry 4.0 as a pathway to a Sustainable Innovation and Inclusive Industrial Technological Development

Summary

This manuscript is a research article focusing on achieving an inclusive pathway towards a sustainable innovative pathway based on the latest technological advancements to enable sustainable design in construction 4.0 (concept based on Industry 4.0). In this study, maintenance, planning and construction of structures has been mainly focused. Therefore, the author tried to collect from different stakeholders of construction industry. However, this study lacks in many aspects from the novelty, incomplete methodology, and main conclusions. English is pathetically poor in this manuscript. English grammar is highly required for this manuscript. Some worth considering comments are:

Comments

  1. This study mainly focusing on the pathway development for the adoption of construction 4.0 in construction sector, no such a clear pathway has been defined till the end of the study. Therefore, major flaws are present between results and defined objectives of the study.
  2. This study lacks in novelty, which has not been clearly explained in this manuscript.
  1. Objectives are not clearly stated, overlap can be found in the defined objectives of this study Objective 2 and 5 are
  2. a qualitative or quantitative study? Mehtodology is incomplete in terms of data collection, study type (qualitative or quantitative)
  3. Population size of 200 construction firms (any idea how many employees work in these firms)? Ambiguity in the sample size with respect to the population size. First similar in meaning, similarly, objective 3 and 6 are overlapping in deeper aspect.
  4. Study does not addresses the objective 4: Investigate the social and economic implication of C4.0 innovations in industrial development
  5. Results are not in align with the defined objectives
  6. How data was collected, not clearly stated in methodology. Was itauthors are considering firms and in the samples size they stated the 150 respondents.
  7. Table 2 how relative agreement index was calculated and which data was used to calculate it?
  8. How ranking was assigned? In table 2
  9. Line # 344 table 5a is missing in the manuscript
  10. Conclusion section is missing in this manuscript.
  11. In tbale 3 hinderance parameters: government policy has been defined as first policy but it can vary from country to country (it would be good if authors mention the country of this study) or from where the data has been collected)

Suggestions

  1. Line # 6 Scene meaning?
  2. Line # 73 stainable ??
  3. Line # 74 goal 9 and 11 of what?
  4. Line #251 (200 firms or persons were contacted)
  5. Line # 315 it should be table 3 instead of table 2
  6. None consistency is present in the use of abbreviations.

Round 2

Reviewer 1 Report

In the revised version authors gave additional insights into their research and also acted upon given comments and suggestions, and gave all required clarifications. Overall, I believe that the article provides valuable content to the present body-of-knowledge.

Reviewer 2 Report

Appreciated work done in review